# Variations in Salivary Stress Biomarkers and Their Relationship with Anxiety, Self-Efficacy and Sleeping Quality in Emergency Health Care Professionals

**DOI:** 10.3390/ijerph18179277

**Published:** 2021-09-02

**Authors:** Daniel Pérez-Valdecantos, Alberto Caballero-García, Teodosia del Castillo-Sanz, Hugo J. Bello, Enrique Roche, Alba Roche, Alfredo Córdova

**Affiliations:** 1Departamento de Bioquímica, Campus Universitario “Los Pajaritos”, Biología Molecular y Fisiología, Facultad de Ciencias de la Salud, GIR de “Ejercicio Físico y Envejecimiento”, Universidad de Valladolid, 42004 Soria, Spain; danielperezvaldecantos@gmail.com; 2Departamento de Anatomía y Radiología, Facultad de Ciencias de la Salud, GIR de “Ejercicio Físico y Envejecimiento”, Campus Universitario “Los Pajaritos”, Universidad Valladolid, 42004 Soria, Spain; alberto.caballero@uva.es; 3Gerencia de Emergencias Sanitarias de Castilla y León, UME Soria, Hospital Virgen del Mirón, 42005 Soria, Spain; tcastillosanz@gmail.com; 4Departamento de Matemáticas, Escuela de Ingeniería de la Industria Forestal, Agronómica y de la Bioenergía, GIR de “Ejercicio Físico y Envejecimiento”, Campus Universitario “Los Pajaritos”, Universidad de Valladolid, 42004 Soria, Spain; hjbello.wk@gmail.com; 5Department of Applied Biology-Nutrition, Institute of Bioengineering, Miguel Hernández University, 03202 Elche, Spain; eroche@umh.es (E.R.); alba.1078@gmail.com (A.R.); 6Alicante Institute for Health and Biomedical Research (ISABIAL Foundation), 03010 Alicante, Spain; 7CIBER Fisiopatología de la Obesidad y Nutrición (CIBEROBN), Instituto de Salud Carlos III, 28029 Madrid, Spain

**Keywords:** amylase, anxiety, DHEA, emergencies, healthcare professional, self-efficacy, sleeping quality, stress

## Abstract

Hospital healthcare workers of emergency departments (EDs) face a variety of occupational stressors on a daily basis. We have recently published that emergency professionals display increased salivary levels of α-amylase and dehydroepiandrosterone during the working day. The pattern of these markers may suggest a counteracting mechanism of dehydroepiandrosterone against the stress reflected by amylase increases. In order to verify this hypothesis, we have analysed different psychological aspects in the same group of healthcare professionals through different tests related to behaviours resulting from stress. These include the state-trait anxiety inventory, the self-efficacy test and the sleeping quality questionnaire. The tests were provided at the beginning of the working day and collected at the end. STAI scores (trait and state) were indicative of no anxiety. Self-efficacy scores were considered optimal, as well as those from the sleeping quality questionnaire. This is supported by the modest correlation between STAI scores and salivary levels of α-amylase and dehydroepiandrosterone. In conclusion, the emergency professionals of the studied hospitals seem to have adequate work management. Altogether it means that the stress generated during work performance is controlled, allowing a correct adaptation to the demanding situations undergone in emergency departments.

## 1. Introduction

Work stress can appear as a behavioural and physiological reaction to harmful aspects of professional tasks, contributing to developing negative psychological and physiological outcomes. Stress leads to a state of imbalance in the system of variables that link people to their environment [1]. Based on this definition, work stress is a multiple process linked to personality characteristics, coping processes and positive and negative work experiences [2]. In this context, health professionals from emergency departments (EDs) face complex tasks on a daily basis that results in a high emotional involvement [3,4]. This is due to particular aspects of work management in EDs.

Anxiety levels (negative stress) have been reported to be higher in health professionals than in the general population. This has been attributed the high psychological demand of frequent night work, with few hours of sleep and gruelling workloads, among other causes [5]. High and maintained levels of stress are involved in cardiovascular disease, increased susceptibility to infections, physiological disorders and mental illnesses, affecting job performance in health professionals [3,4].

Stress implies an activation of the hypothalamic–pituitary–adrenal (HPA) axis and the autonomous nervous system (ANS), triggering the neurohormonal processes that take part of the body’s psychosomatic response to the specific stress situation [6]. Activation of the HPA axis is subsequent to activation of the sympathetic–adreno–medullar (SAM) system. Once a stressor is perceived, the release of the corticotropin-releasing hormone (CRH) is stimulated. Subsequently, the CRH stimulates the pituitary gland, favouring the release of adrenocorticotropin (ACTH), which stimulates the adrenal cortex to release glucocorticoids, which include cortisol, androstenedione and dehydroepiandrosterone (DHEA). Altogether CRH, ACTH, DHEA-sulphate (DHEA-s) and the non-sulphate form (DHEA) are hormonal biomarkers of the HPA axis. These hormones can reach saliva by intracellular passive diffusion, presenting a good correlation with circulating blood levels [7]. In addition, the rapid activation of the SAM system leads to high concentrations of epinephrine and norepinephrine in blood. Salivary glands have a high number of β-adrenergic receptors that, stimulated by norepinephrine, favour the secretion of salivary α-amylase [8].

The advantage of salivary α-amylase over cortisol is the faster release (10–15 min) after stimulation. Meanwhile the release of cortisol occurs 25 to 30 min after stimulus appearance [9]. On the other hand, salivary DHEA appears as a response from the stimulation of the HPA. It seems that DHEA exerts a beneficial effect as an anti-stress messenger, antagonizing the cortisol action. In a recent study, the authors observed that DHEA plays an instrumental role in anxiety management during the pubertal period [10]. However, studies regarding the role of DHEA in adults undergoing stress situations are still lacking.

### 1.1. Stress and Anxiety

It is well established that people suffering from intense stress use to undergo anxiety. In the context of medical EDs, stress can affect individual and team performance when treating patients, with a relationship between acute stress and established task execution [11,12]. In this context, anxiety appears in individuals who do not have the tools to adequately cope with stress. Anxiety needs the adaptive mechanisms of stress to develop. However, the long-term consequences undergone by an anxious subject could represent a serious problem for their health and well-being [13]. Therefore, anxiety disorders are varied combinations of physical and mental manifestations, not attributable to a real danger. They used to occur in the form of a crisis or as a persistent state. Currently, it is accepted that there is a psychogenic anxiety and an endogenous anxiety [14], conceived in the context of personality in terms of trait and state [15,16]. Altogether, anxiety is a desirable reaction, because it tends to help the individual generating alternative responses to find a better outcome, unless it is experienced with an excessive intensity, frequency or persistency. In these cases, anxiety would lead to interferences in people’s daily lives, generating deep discomfort and possible disorders [17]. When the stressful situation is persistent (chronic stress), it is possible to observe alterations in the function of the neurobiological stress circuit, such as a mismatch in the SAM axis. This situation results in an association of catecholamine production with anxiety states [18,19,20]. On the other hand, Ruotsalainen et al. [21] indicate that, in response to the ever-changing demands of their work, healthcare professionals could learn to mitigate the negative consequences of anxiety in situations of acute stress.

Taking into account physiological responses, Boudarene et al. [22] identified three types of stress responses: (a) situations with no anxiety (psychological silence) and no increase in cortisol levels (biological silence), (b) high levels of anxiety (high emotional reaction) with no biological manifestations (biological silence), and (c) high levels of anxiety as well as plasma cortisol. In a recent review [23], it was concluded that young resident physicians receive a progressively increasing workload parallel to increasing levels of stress, reporting that 20–30% of resident physicians may experience relevant levels of anxiety. This situation affects quality of life and health of professionals resulting in less self-efficacy and sleeping quality [24]. Sleep is necessary for the optimal setup of psychological and organic functions and is altered in numerous situations. Among them, anxiety is possibly one of the most disturbing [25,26]. The impact of insomnia on quality of life is manifested in daytime sleepiness, decreased performance at work, changes in character with impaired interpersonal relationships and increased risk of accidents [25,26]. In addition, sleep problems are risk factors for psychological disorders such as depression and suicide [27].

### 1.2. Stress, Sleep Quality and Self-Efficacy

It has been reported that high levels of anxiety and depression associated with short sleep patterns and poor quality of sleep, affect mental and work performance, compromising the safety of the patient and the doctor themself [28]. It has been recognised those subjects who are more satisfied with their sleep in the long run have less anxiety compared to those who are not [29]. Nevertheless, it is true that certain doses of anxiety promote positive and task-focused performance, compared with longer and recurrent anxiety states [30,31]. In this context, positive scores in self-efficacy results in better job performance, creating a state of emotional tranquillity focused on good work [32]. Contreras et al. [33] reported that academic performance is directly associated with self-efficacy with no anxiety. People with elevated self-efficacy and control perception have lower cardiovascular reactivity [34]. On the contrary, subjects who underestimate their abilities, increase physiological activation and desist when task execution is complicated [35]. In addition, it has been reported that greater self-efficacy may be associated with positive healthy behaviours such as physical activity performance in adolescents [36].

Our hypothesis is that well-prepared and trained ED professionals can cope optimally with the stress situations they have to face during the working day. This control can be monitored by determining salivary stress biomarkers where anti-stress hormones, such as DHEA, seem to counteract the action of stress hormones, such as cortisol and α-amylase. These changes can be reflected in the answers provided by these professionals to different tests that analyse different psychological aspects related to behaviours resulting from stress, such as anxiety, self-efficacy and sleeping quality (Figure 1).

In a previous report, we determined salivary stress related biomarkers, such as cortisol, α-amylase and DHEA, in health professionals working in hospital EDs. We have observed that cortisol and DHEA levels were high early in the morning (8:00 h) and decreased through the working day, reaching the minimal value at the end of the day (24:00 h). On the other hand, α-amylase reached the peak at 15:00 h, decreasing thereafter. In this context, it is generally assumed that anxiety, sleeping quality and self-efficacy are modulated by different stress situations. However, very few studies have addressed all these parameters in health professionals working in EDs. The objective of the present report is to study how this particular pattern of stress biomarkers (particularly α-amylase and DHEA [37]) observed in emergency professionals, could influence anxiety, self-efficacy and sleeping quality. To this end, different specific tests were filled by these professionals the same day that saliva samples were collected to determine the different hormones [38]. The obtained results could give key information regarding the influence of these stress responses in working performance and professional actions.

## 2. Materials and Methods

An analytical, cross-sectional and descriptive study was carried out during the months of July and August 2019 in the EDs of two public Spanish hospitals: Hospital Clínico Universitario de Valladolid (HCUV) (third level) and Hospital Santa Bárbara de Soria (HSBS) (second level). The total staff in HCUV includes 502 medical doctors (28 working in the ED) and 865 nurses (82 working in the ED). The total staff in HSBS includes 187 medical doctors (18 working in the ED) and 269 nurses (22 working in the ED). The project was approved by the Ethics Committee for Drug Research of the Burgos Health Division with reference CEIC-1984 (24 July 2018). A total of 97 professionals participated in the study: 59 nurses (49 women and 10 men) and 38 medical doctors (28 women and 10 men) (Table 1). The mean age of medical doctors and nurses of both hospitals was 40.2 and 39.8 years, respectively. No significant differences were found compared to the participants in the study. As inclusion criteria, all participants have normal and stable lifestyle and family habits, regarding family tasks, physical activity and nutritional habits. In general, a stable lifestyle is considered to be an existence that facilitates physical, psychological and social well-being and maintains favourable conditions to preserve and promote its development. Otherwise said, participants have social, physical activity and nutritional habits that allow maintaining the standards of a healthy life. In addition, participants were free from any mental or physical pathology, especially endocrine diseases. Participation in the study was voluntary (all participants signed a written consent) and anonymous. Participants were not funded and were informed that they could withdraw from the study at any time with no explanations. See more details in [38].

For the assessment of physiological stress response, salivary measurements of α-amylase and DHEA were performed from samples obtained in 4 moments of the day: 8 h, 12 h, 15 h and 24 h. Saliva samples were obtained using the Salivette commercial kit^®^ (Sarstedt International, Nombrecht, Germany). Subsequent detection was performed by Elisa immunoassay with the following references: SALV-2930 DRG for cortisol, EIA-5836 DRG for α-amylase and SALV-3012 DRG for DHEA. Participants were advised to avoid eating or smoking 60 min before saliva collection. Samples were maintained on ice and then at −20 °C. At the moment of analysis, samples were thawed and centrifuged at 3000 rpm for 5 min at 4 °C. ELISA immunoassay was performed for cortisol (SALV-2930 DRG, Marburg, Germany), α-amylase (EIA-5836 DRG, Marburg, Germany) and DHEA (SLV3012 DRG, Marburg, Germany) detection. Results have been published previously [38]. The day that saliva samples were collected to determine the different hormones, the psychologists of the team passed different specific tests.

The trait-state anxiety inventory (STAI) was used to measure anxiety. This tool has two self-assessment scales to measure two independent concepts of anxiety: state and trait. Both the status scale and the trait scale have 20 items each, which are scored on a Likert scale with 4 response options (from 0 to 3). Options to score in anxiety/state scale are: 0 (nothing), 1 (something), 2 (enough) and 3 (a lot). Options to score in anxiety/trait scale are: 0 (never), 1 (sometimes), 2 (often) and 3 (always). The questionnaire has a good internal consistency in the Spanish adaptation, between 0.9 and 0.93 in anxiety/state and between 0.84 and 0.87 in anxiety/trait [16].

The self-efficacy test refers to the feelings of own ability, sensitivity and caution. The general self-efficacy scale [39] assesses the stable feeling of personal capability to effectively handle a wide variety of stressful situations. The questionnaire consists of 10 items encompassing the dimension of self-efficacy at work and evaluating people’s stable belief in the ability to properly handle a wide range of stressors. The minimum and maximum score on the scale is 10 and 40, respectively.

Finally, the COS (from Spanish “Cuestionario de Oviedo del Sueño”/Oviedo’s Sleep Questionnaire) was used to assess sleep. The COS [40] consists of 15 items that allow obtaining information regarding the sleep and wakefulness rhythms of the person. The questionnaire consists of 3 subscales for: 1 item for subjective sleep (scale = 1–7 points scored on a Likert scale); 9 items for insomnia, such as severity, sleep latency, duration and efficacy; daytime dysfunction (scale = 1–5 points scored on a Likert scale); 3 items for hypersomnia (scale = 1–5 points scored on a Likert scale). The remaining 2 items provide information on the use of sleep medications or the presence of adverse events during sleep. The subscale of insomnia ranges from 9 to 45. Higher scores indicate a higher severity of insomnia. High COS score indicates high levels of sleep disorders. The COS shows high reliability (α Cronbach’s = 0.76). All questionnaires were provided early in the morning at home to each participant. After a short explanation, questionnaires were completed by each participant in a calm environment, usually after work at home. Then questionnaires were picked up at the end of the working day.

Data were analysed with the R, R-Studio and Python software package (Pandas, Numpy). Quantitative variables were expressed using 95% confidence interval for the mean, obtained using non-parametric bootstrap analysis (necessary due to non-normality). Histograms and QQ-plots were used to assess the non-normality of our data.

## 3. Results

In a previous report, we have published the variations of α-amylase and DHEA through the working day in professionals working in hospital EDs. Salivary levels were determined at 8h, 12 h, 15 h and 24 h. α-amylase levels were low at 8–12 h reaching a significant peak at 15 h (302.35 ± 35.55 U/mL) and decreasing thereafter at 24 h. On the other hand, the levels of DHEA were significantly high at 8 h (301.85 ± 44.55 pg/mL), decreasing the rest of the day at 12, 15 and 24 h [38]. No differences were found when comparing men vs. women.

Table 2 shows the scores obtained from the STAI questionnaire regarding state and trait concepts. No significant differences were observed when comparing gender, professional status and hospitals (Table 2). However, α-amylase was significantly higher in medical doctors compared to nurses only at 15 h. In addition, α-amylase was significantly higher in the HCUV compared to the HSBS only at 8 and 12 h [38].

Since anxiety does not seem to be affected, we wanted to see if other psychological variables present changes in emergency health professionals. Table 3 shows the scores obtained from the self-efficacy test. As well as that observed in the STAI, no significant differences were observed when comparing gender, professional status and hospitals.

The STAI questionnaire is a psychological tool to determine anxiety. In addition, α-amylase is a biomarker that increases in saliva in response to stress. Therefore, we wanted to see if a correlation exists between the different salivary α-amylase and DHEA levels through the workday and the STAI scores. The correlation chart in Figure 2 indicates that the correlation between anxiety and α-amylase is low. A Pearson correlation test returns no significant *p*-values (>0.05) when comparing the STAI variables (state and trait) with α-amylase variables (24, 15, 12 and 8 h). However, it returns significant *p*-values when comparing the α-amylase variables with each other. With respect to DHEA and STAI (Figure 3), a Pearson correlation test returns non-significant *p*-values (>0.05) when comparing the STAI variables (state and trait) with the DHEA variables (24, 15, 12 and 8 h). However, it returns significant *p*-values when comparing the DHEA variables with each other.

Since stress can affect sleep, we analysed this variable in the health professionals studied. Table 4 shows the scores obtained in the COS questionnaire regarding three aspects related to sleep: subjective satisfaction of sleep, insomnia and hypersomnia. The scores are low, indicating acceptable levels of sleep. No significant differences were observed regarding gender, professional status (medical doctors and nurses) and hospitals.

Table 5 shows comparisons and their respective *p*-values for all variables analysed for each shift (morning vs. afternoon), for each hospital (HSB vs. HCUV) and in the two professional categories (Physician vs. Nurses). Comparisons were made with a Wilcoxon ranked sum test.

## 4. Discussion

The objective of the present report is to study how the pattern of salivary stress biomarkers (particularly α-amylase and DHEA) observed in emergency professionals and published previously [38] could influence anxiety, self-efficacy and sleeping quality. To this end, different specific tests were filled by healthcare professionals the same day that saliva samples were collected [38]. The obtained results could give key information regarding the influence of these stress responses in working performance and professional actions.

In our previous published report [38], the most relevant observation is that the stress generated throughout the working day, is more dependent on the response of the sympathetic–adreno–medullar system. Biomarkers, such as salivary α-amylase, are objective indicators of chronic stress, but they can also respond to acute psychological stress [41,42]. Therefore, the elevation of salivary α-amylase concentrations could be influenced by processes other than chronic stress [41,42]. Our previous results regarding salivary α-amylase indicate that this biomarker was within normal ranges. Although it increases throughout the working day, this does not mean the appearance of remarkable biological alterations. Therefore, we could say that this is the response of acute stress throughout the working day. This occurs because the adrenal marrow system (amylase-related) responds faster than the cortex system (cortisol-related). Consequently, increased levels of salivary α-amylase are associated to increased sympathetic nervous system activity.

On the other hand, Chatterton et al. [8] investigated stress-induced salivary α-amylase increases and found a positive correlation between biological parameters (epinephrine and norepinephrine) only when stress was induced by a physical factor, but not when stress was induced by a psychological factor. In this context, Rohleder et al. [41] applied a standardised stress factor (TSST or Trier Social Stress Test), which is known to induce strong physiological responses. They found a significant correlation between stress-induced salivary α-amylase increases and plasma norepinephrine levels. These data are consistent with previous studies that like in ours [38], have observed significantly reduced cortisol levels over time [43,44]. This suggests that, despite the increases in salivary α-amylase, there is a decrease in accumulated stress during the working day in health professionals of EDs [43]. Some authors indicate that increased salivary α-amylase level may pose a faster reaction to stress than cortisol, suggesting that it is a better stress biomarker. According to Nater et al. [42], salivary α-amylase has a pattern of daytime secretion, indicating that it decreases sharply 60 min after waking up and increases consistently throughout the day. In a study in adolescents, low salivary α-amylase levels have been reported in the morning, increasing over the course of the day [45]. These results are consistent with the observations obtained by Thoma et al. [46]. Similar studies concluded that there is a clear association between the response to acute stress and the release of salivary α-amylase. Finally, some studies have compared a real fact with a simulation [43]. They have observed that the simulation produces increases in α-amylase similar to observations based on real work actions.

### 4.1. Stress and Anxiety in the Studied ED Professionals

Regarding anxiety, the anxiety state is defined by the authors who created the test [39] as a transient emotional condition of the organism, characterised by subjective feelings of tension and apprehension. However, the anxiety trait is defined as a stable anxious propensity that makes people and situations perceived as threatening. In the analysis of our data, we do not see variations between the two situations (trait and state). In addition, STAI scores were in levels that can be considered good, both in men and women, as well as in medical doctors and nurses and in both hospitals (HCUV and NSBS). Altogether, this suggests that healthcare professionals are not affected by the situations they have to live in in the EDs. We could say that their work situation is considered normal, suggesting an allostatic adaptation [47,48]. According to the allostatic model, continuous or prolonged repeated stress events lead to a homeostatic restart to redefine the conditions necessary to maintain the homeostasis. In this context, studies performed in humans and animal models reported a decrease in the stress response when a sustained repetition occurs [47,48,49].

In light of our results and taking into account the limitations that may arise from sample size, we believe that the studied participants from the EDs of the hospitals have adequate management of work. Taking into account the central role of ANS in numerous pathological states, reflected by changes in salivary α-amylase, this may suggest that this could be a key element in anxiety-related disorders. In addition, salivary α-amylase assessment can complement other determinations of the autonomous system response, such as heart rate variability [50,51,52].

Interestingly, when comparing our STAI state data with other previous studies, our findings are lower, both in the total ranges, by gender, profession and hospital. A first explanation could be attributed to the high level of awareness and preparation of emergency health professionals studied in this work. As indicated by Ellershaw et al. [49]: “scrupulousness is the strongest driving force in the performance of the job role”.

Nevertheless, we cannot ignore a biological response. In our previous publication [38], we have observed that DHEA levels decrease, and this could be related to the allostatic adaptation. We could speculate that low DHEA levels could indicate adrenal depletion/allostasis in the studied individuals. This is pointing to HPA axis-dependent hypoactivity. However, we believe that this is not due to exhaustion but mostly to adaptation, because when analysing the response from a psychological point of view (STAI), we noted that the response of the participants, both men and women, was maintained in similar trait/state values. Other authors studying groups of adolescents who had suffered from traumatic stress, observed a decrease in DHEA levels in correlation with adrenal exhaustion [50,51]. Trickett et al. [52] support the theory of a change from hyper to hyposensitivity to glucocorticoids and hypercortisolism in response to years of child abuse.

Viljoen et al. [53] indicate that DHEA has significant predictive power in the context of propensity to anxiety and propose it as a biomarker to monitor progressive changes related to anxiety in the HPA axis. In this context, we observed that the levels of cortisol decreased together with the anti-stress hormone DHEA as well [54,55]. However, we have not seen variations in scores from the STAI. We hypothesise that the levels of the stress biomarkers were in very acceptable ranges, indicating that the studied individuals are capable to deal with stress situations at work. In addition, a pilot study indicated that morning serum concentrations of DHEA were positively correlated with the anxiety severity subscale score [56]. This leads us to believe that DHEA actually has an anti-stress effect. In this regard, it has been seen that DHEA-s has anxiolytic and antidepressant effects through antagonistic activity on N-methyl D-aspartate receptors [57].

### 4.2. Stress and Self-Efficacy in the Studied ED Professionals

The role performed by DHEA is confirmed by the self-efficacy test that determines the self-confidence to be able to follow a proposed behaviour. This is a key aspect to consider in relationship to the development of a task in an efficient and adequate way. In fact, self-efficacy may decrease the burden of stress and anxiety. Our data shows a high level of self-efficacy, which plays a key role in transforming emotions at work. In this sense, Castaño et al. [58] conducted a study with doctors of EDs, confirming that self-efficacy is a key aspect in emotional control. These observations have also been confirmed by others [59,60]. The results from our work show a high rate of self-efficacy (29 out of 40). Another study conducted by Mo et al. [61] observed that nurses have a high self-efficacy score of >20. Correlation analysis also indicated that self-efficacy was negatively correlated with anxiety. Furthermore, in a longitudinal study with an adult population in China, general self-efficacy was also negatively correlated with stress and anxiety [62]. In our study, self-efficacy presented a high negative correlation with both state and trait anxiety determined with STAI (not shown). Therefore, we can speculate that high expectations of self-efficacy would enable participants to deal effectively with problems that could appear during work execution. In this way, this can result in a better state of health for participants, or even to be motivated to perform healthier lifestyle habits, allowing to efficiently cope with working tasks [63].

Altogether, self-efficacy is important to improve work efficiency, job motivation and work attitudes [64,65]. People with a strong sense of self-efficacy have the courage to overcome difficulties and exhibit good emotional and behavioural states. Therefore, nursing managers should pay attention to cultivate self-efficacy, improving self-confidence in stress and reducing anxiety. However, high self-efficacy is not a sufficient condition for proper execution. The skills, material resources and incentives necessary to act [66] must be present as well.

### 4.3. Stress and Sleep Quality in the Studied ED Professionals

Moreover, the results concerning low anxiety and high self-efficacy are related to the good quality of sleep perceived by healthcare professionals. Sleep is regulated by homeostatic and circadian rhythms that are independent but influence the amount (sleep hours) and timing (moment of sleep) of an individual under normal conditions. The regular sleep rhythm is influenced by events that occur during wakefulness such as stress [67]. In stressed people, anxiety is often a common ground. According to Selye, this results in a psychosomatic disorder that begins with affecting brain function and has an impact on various organs [68].

Rotenberg and Arshavsky [69] consider that changes in sleep are not caused by stress, but by the type of behavioural reaction to emotional stress. It seems that the work execution could cause sleep disorders, resulting in certain degree of daytime sleepiness. This situation has an effect on mood leading to irritability, anxiety, tension, fatigue, among others, influencing work activities [70]. In addition, sleep disorders in anxious patients can be quite severe and long-lasting, causing difficulty to relax or worrying about the day’s problems when they go to bed [71].

### 4.4. Limitations of the Study

Additional tests can complete the data obtained. In this regard, external aspects, outside the workplace, such as lifestyle and family habits must be also taken into account. In this particular study, all participants declared stable lifestyle and family habits [38]. This aspect can be considered as a third dimension to manage health effects due to stress. In the 1970s, two dimensions were identified that had effects on stress-related health status: psychological demands and control capacity [72]. It was later, when a third variable was included: social support [73]. This variable refers to the social climate in the workplace in relation to peers and superiors, and can be defined as a robust social network, a significant number of social contacts or the possibility of expressing concerns and intimacies [73]. Social and family support increases the ability to cope with a maintained stressful situation, so it is considered a buffer of stress associated with the job [73]. Maintaining good personal and professional relationships between peers, participating in professional forums, maintaining stable family relations, friendships and support groups, could contribute to the prevention and appearance of different pathologies derived from stress [74]. Under this perspective, it would be worth noting the importance of measuring this variable in future research since it seems to influence work stress as a buffer. This could explain in part the low levels of anxiety observed in this study.

On the other hand, other limitations presented by this study could be the measurement time. According to Steiler and Rosnet [75], when measuring work stress two central aspects should be considered: (a) stress is individual and depends on how workers adjust and cope with work situations; (b) stress is an ongoing process and may vary over time, so a single measurement is not enough. Since work stress is changing, it would be interesting to extend the measurement time of subjects’ stress levels in order to obtain, in the future, a longitudinal study.

Another limitation of the study is that the sample size of this study (n = 97) was perhaps smaller than required for a larger study and clearer conclusions. This limitation is due to the lack of access to a larger sample, although we believe that the sample size achieved may be sufficient for the population studied (HSB of Soria).

Although sleep disorders and anxious mood are the predominant manifestations of anxiety, in our study we did not see alterations that could compromise psychological integrity of participants. According to our data, the quality of sleep was good, with low anxiety and high efficiency, which was not influenced by the gender or profession in the EDs of the two hospitals studied (HCUV and HSBS). Anxiety, self-efficacy and sleep quality in the studied emergency health professionals are at levels considered very good, which makes the performance of their tasks efficient. This can contribute to a better patient’s care. In conclusion, the emergency professionals of the studied hospitals have adequate work management, and the stress controlled during work performance can allow for an optimal adaptation to the demanding situations undergone in EDs.

## Figures and Tables

**Figure 1 ijerph-18-09277-f001:**
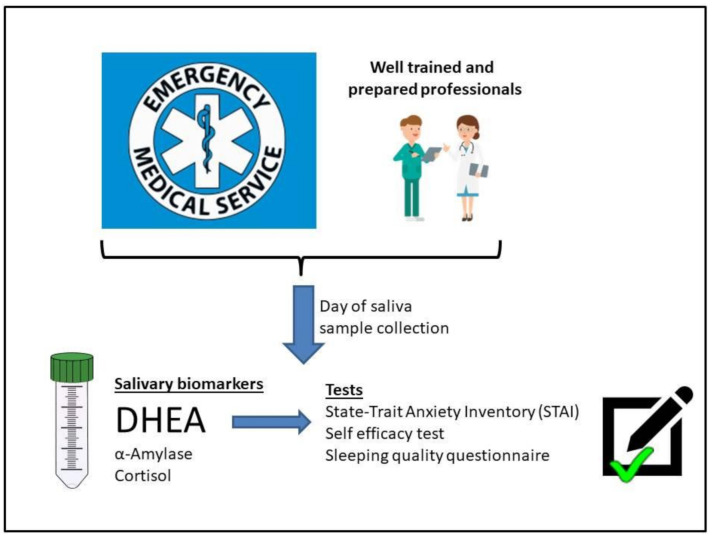
Scheme of the hypothesis presented in this report (images obtained from Wikimedia Commons).

**Figure 2 ijerph-18-09277-f002:**
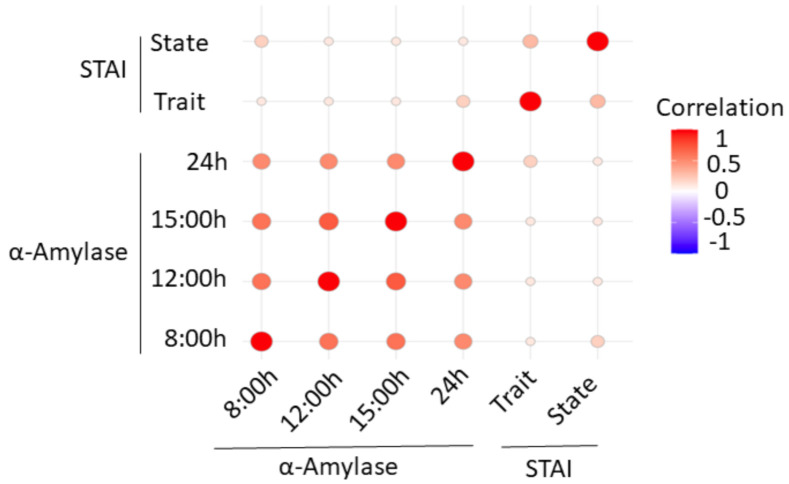
Correlation chart of α-amylase levels determined at 8 h (197.6 ± 37.1 U/mL), 12 h (283.8 ± 35.3 U/mL), 15 h (302.4 ± 35.6 U/mL), 24 h (239.7 ± 29.2 U/mL) [38] and STAI scores.

**Figure 3 ijerph-18-09277-f003:**
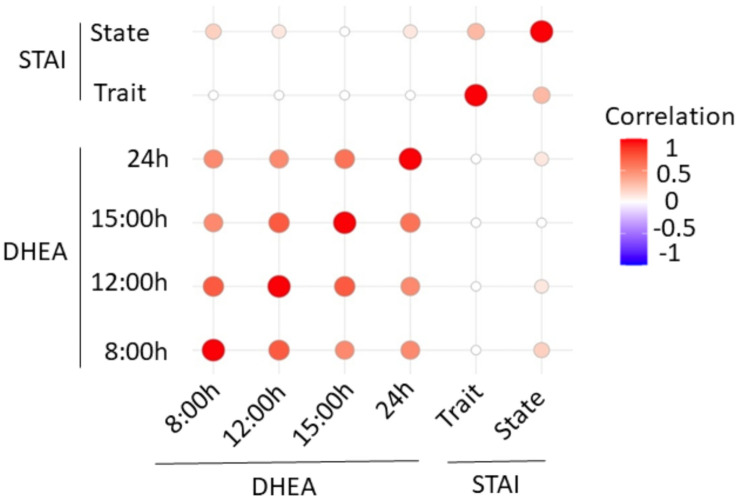
Correlation chart of DHEA levels determined at 8 h (301.9 ± 44.6 pg/mL), 12 h (250.9 ± 35.3 pg/mL), 15 h (235.7 ± 33.6 pg/mL), 24 h (221.4 ± 30.3 pg/mL) [38] and STAI scores.

**Table 1 ijerph-18-09277-t001:** Professional groups and mean age ± SD of participants.

Professionals	*n*	Age
Total	97	39.5 ± 12.1 (women) 39.9 ± 15.2 (men)
Nurses	59	39.0 ± 13.2
Medical Doctors	38	39.6 ± 13.5
HCUV	45	34.7 ± 9.7
HSBS	52	42.4 ± 12.5

Abbreviations used: HCUV, Hospital Clínico Universitario de Valladolid; HSBS, Hospital Santa Bárbara de Soria.

**Table 2 ijerph-18-09277-t002:** STAI scores (state and trait) regarding gender, professional status (nurses and medical doctors) and hospitals.

	STAI State	STAI Trait
Women	25.05 ± 1.15	24.69 ± 1.23
Men	25.67 ± 1.77	24.31 ± 1.93
Nurses	25.61 ± 0.92	24.94 ± 1.18
Medical Doctors	24.53 ± 2.03	24.27 ± 1.89
HCUV	25.04 ± 1.28	24.89 ± 1.33
HSBS	24.66 ± 1.68	24.02 ± 1.98
TOTAL	25.18 ± 0.99	24.73 ± 1.07

Results are expressed using 95% confidence interval for the mean, obtained using non-parametric bootstrap analysis (necessary due to non-normality). Abbreviations used: HCUV, Hospital Clínico Universitario de Valladolid; HSBS, Hospital Santa Bárbara de Soria.

**Table 3 ijerph-18-09277-t003:** Self-efficacy scores regarding gender, professional status (nurses and medical doctors) and hospitals.

	Self-Efficacy Scores
Women	29.07 ± 0.99
Men	30.02 ± 2.03
Nurses	29.12 ± 0.92
Medical Doctors	29.54 ± 1.79
HCUV	30.22 ± 0.85
HSBS	28.26 ± 1.66
TOTAL	29.30 ± 1.01

Results are expressed using 95% confidence interval for the mean, obtained using non-parametric bootstrap analysis (necessary due to non-normality). Abbreviations used: HCUV, Hospital Clínico Universitario de Valladolid; HSBS, Hospital Santa Bárbara de Soria.

**Table 4 ijerph-18-09277-t004:** COS scores regarding gender, professional status (nurses and medical doctors) and hospitals.

	Subjetive Satisfaction of Sleep	Insomnia	Hypersomnia
Women	4.13 ± 1.36	19.95 ± 7.23	5.68 ± 2.66
Men	4.15 ± 1.90	17.33 ± 8.09	5.74 ± 3.60
Nurses	4.16 ± 1.47	19.47 ± 7.56	5.46 ± 2.78
Medical Doctors	3.98 ± 1.65	18.83 ± 7.99	5.97 ± 3.17
HCUV	4.22 ± 1.33	19.27 ± 7.04	5.93 ± 2.55
HSBS	4.18 ± 1.60	19.55 ± 7.76	5.69 ± 3.13
TOTAL	4.10 ± 1.54	19.13 ± 7.69	5.64 ± 2.55

Results are expressed using 95% confidence interval for the mean, obtained using non-parametric bootstrap analysis (necessary due to non-normality). Abbreviations used: HCUV, Hospital Clínico Universitario de Valladolid; HSBS, Hospital Santa Bárbara de Soria.

**Table 5 ijerph-18-09277-t005:** *p*-values for all comparisons for the variables, moment, category and hospital analysed.

	Shift (Morning vs. Afternoon)	Hospital (HSB vs. HCUV)	Category (Phycisian vs. Nurses)
DHEA 8 h	>0.05	>0.05	>0.05
DHEA 12 h	>0.05	>0.05	>0.05
DHEA 15 h	0.04575	0.005375	>0.05
DHEA 24 h	0.03665	0.03187	>0.05
α-Amylase 8 h	>0.05	0.004259	>0.05
α-Amylase 12 h	>0.05	0.0295	0.05578
α-Amylase 15 h	0.01023	>0.05	0.003534
α-Amylase 24 h	0.03567	>0.05	>0.05
STAI State	0.02403	>0.05	>0.05
STAI Trait	>0.05	>0.05	>0.05

## Data Availability

The data that support the findings of this study are available from the corresponding author, upon reasonable request.

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
