# Peer review of "Variations in Salivary Stress Biomarkers and Their Relationship with Anxiety, Self-Efficacy and Sleeping Quality in Emergency Health Care Professionals"

_ijerph, 2021, doi:10.3390/ijerph18179277_

Round 1
Reviewer 1 Report
I think that all major concerns have been addressed now, and after a final English grammar and spelling check I think the manuscript seems eligible for publication to a broader audience.
Author Response
No reviewer comments
Reviewer 2 Report
I have read the paper of Pérez-Valdecantos et al. with interest. The authors took the interesting topic of variations in salivary stress biomarkers and their relationship with anxiety, self-efficacy, and sleeping quality in emergency health care professionals. Although it is a good written, comprehensive review, there are some elements, which should be corrected before final publication:
Major:
- The hypothesis of the study should be added to the introduction
- You provided a very comprehensive introduction. In my opinion, you should provide the figure summarizing the introduction.
- line 158: 97 professionals -> the missing information about a total number of professionals (FROM ... working in these EDs). How did you choose them from others?
- The meaning of "normal and stable lifestyle and family habits" is puzzling. It should be better described. How was it checked? The information about the mean age of nurses/medical doctors/HCUV/HSBS should be provided with information if there are any significant differences between them and the chosen group.
-Line 211-213: The statistical analysis is poorly described. Did you test the normality of obtained data? If yes which test did you use? If no, why did you provided mean instead of median?
Minor:
- Line 163: "In addition, participants were free from any mental or physical pathology as well as endocrine-type diseases." - endocrine-type diseases are physical pathology. Therefore, it should be stated: In addition, participants were free from any mental or physical pathology, especially endocrine diseases.
- Line 175: The catalog number of the used kit should be provided.
Author Response
Reviewer 2 (3th round)
I have read the paper of Pérez-Valdecantos et al. with interest. The authors took the interesting topic of variations in salivary stress biomarkers and their relationship with anxiety, self-efficacy, and sleeping quality in emergency health care professionals. Although it is a good written, comprehensive review, there are some elements, which should be corrected before final publication:
ANSWER: We appreciate these comments and we hope that the corrected version will be improved with the suggestions made by the Reviewer.
Major:
- The hypothesis of the study should be added to the introduction.
ANSWER: We have included the hypothesis at the end of the Introduction, just before the last paragraph that presents the objective of the work.
- You provided a very comprehensive introduction. In my opinion, you should provide the figure summarizing the introduction.
ANSWER: To complete the Introduction, we have added a figure (Figure 1) that summarizes the main points of the hypothesis.
- Line 158: 97 professionals -> the missing information about a total number of professionals (FROM ... working in these EDs). How did you choose them from others?
ANSWER: At the moment of the study, the medical and nurse staff of both hospitals was as follows: HCUV includes 502 medical doctors (28 working in the ED) and 865 nurses (82 working in the ED). HSBS includes 187 medical doctors (18 working in the ED) and 269 nurses (22 working in the ED). These data have been included in the first paragraph of Materials and Methods. From these, 97 professionals participate in the study. We selected a regular working day. One of selection criteria was that both Hospitals were from University of Valladolid, where the research team was located. The working system is these particular Hospitals is similar and representative of the Spanish Public Health System.
- The meaning of "normal and stable lifestyle and family habits" is puzzling. It should be better described. How was it checked? The information about the mean age of nurses/medical doctors/HCUV/HSBS should be provided with information if there are any significant differences between them and the chosen group.
ANSWER: We asked very general questions about daily routine tasks with the rest of the family, physical activity performed, activities during free time and nutritional habits. The change has been added and explained in the text (first paragraph of Materials and Methods). The mean age of medical staff and nurses of both Hospitals is 40.22 and 39.75 years respectively. No significant differences were found compared to the participants in the study. The information has been included in the text (first paragraph of Materials and Methods).
-Line 211-213: The statistical analysis is poorly described. Did you test the normality of obtained data? If yes which test did you use? If no, why did you provided mean instead of median?
ANSWER: We have defined better the statistical analysis performed in the last paragraph of Materials and Methods. The change was done accordingly in Tables 2-4.
Minor:
- Line 163: "In addition, participants were free from any mental or physical pathology as well as endocrine-type diseases." - endocrine-type diseases are physical pathology. Therefore, it should be stated: In addition, participants were free from any mental or physical pathology, especially endocrine diseases.
ANSWER: The change has been performed accordingly (first paragraph of Materials and Methods).
- Line 175: The catalog number of the used kit should be provided.
ANSWER: The reference number of the kits used for the different hormone determination has been added (second paragraph of Materials and Methods).
Round 2
Reviewer 2 Report
The authors addressed all the comments suficiently improving the manuscript.
This manuscript is a resubmission of an earlier submission. The following is a list of the peer review reports and author responses from that submission.
Round 1
Reviewer 1 Report
Understanding how the stress response system of these professionals responds to the emergency stress is important to improve working conditions and the health of emergency professionals. In this article the authors examined the salivary stress marker levels at different time of the working day and their correlation with anxiety, self-efficacy, and sleep quality in emergency health care professionals. No significant correlation was found, and the authors conclude that stress generated in emergency professional is adapted and controlled.
Concerns
A brief background information about roles of α-Amylase and DHEA in stress response will help readers understanding why selecting these markers.
Measurement methods for α-Amylase and DHEA were missed
For Fig1 and 2, no significance of the correlation coefficient was provided.
Introduction and discussion can be more concise.
Author Response
REVIEWER 1
Understanding how the stress response system of these professionals responds to the emergency stress is important to improve working conditions and the health of emergency professionals. In this article the authors examined the salivary stress marker levels at different time of the working day and their correlation with anxiety, self-efficacy, and sleep quality in emergency health care professionals. No significant correlation was found, and the authors conclude that stress generated in emergency professional is adapted and controlled.
GENERAL COMMENT
Dear Reviewer, we appreciate your suggestions, which will certainly improve the manuscript. Below we try to answer your questions and concerns.
All changes have also been introduced in the manuscript.
Concerns:
A brief background information about roles of α-Amylase and DHEA in stress response will help readers understanding why selecting these markers.
ANSWER: Biomarkers are substances, structures or processes capable of being measured and evaluated as indicators of normal or pathological biological states. A biological marker must be sensitive, objective, specific, stable and quantifiable (Strimbu & Tavel, 2010). Biomarkers of the stress response are conducted immediately by the release of catecholamines mediated by the sympathetic adrenomedular (SAM) axis, and more slowly, by the hypothalamic-pituitary-adrenal (HPA) axis. Both α-amylase and DHEA, are stress biomarkers recognized and used widely by scientists in stress studies. Both biomarkers are easy to measure in saliva by non invasive procedures. In particular, salivary α-amylase level reflects the adrenergic activity secondary to bio-psycho-social stress of the SAM system and, thereby, it is considered a biological marker of psychological stress. In this context, the SAM axis provides an activation response, in a general way to physical and psychological stress, which is modulated by the parasympathetic system. On the other hand, activation of the HPA axis is subsequent to activation of the SAM axis. Once a stressor is perceived, the release of corticotropin-releasing hormone (CRH) is stimulated. Subsequently, CRH stimulates the pituitary gland, which, in turn, releases adrenocorticotropin (ACTH) which stimulates the adrenal cortex to release glucocorticoids. Altogether, CRH, ACTH, dehydroepiandrosterone sulfate (DHEA-s) and its non-sulfate form (DHEA) are hormonal biomarkers of the HPA axis. In this context, Laurent et al. indicated that the cortisol response is typically aligned with the response of other stress systems such as a protective HPA axis component represented by DHEA-s and an autonomic mobilization component represented by α-amylase. The observed multilevel coordination between cortisol and DHEA-s is consistent with a previous evidence of an association between the two adrenal outputs, and may help to understand beneficial impacts of DHEA-s during stress (i.e. as a buffer/repair mechanism protecting against cortisol impact).
- Strimbu, K.; Tavel, J.A. What are biomarkers? Curr Opin HIV AIDS 2010, 5, 463-466.
- Laurent, H.K.; Lucas, T.; Pierce, J.; Goetz, S.; Granger, D. A. Coordination of cortisol response to social evaluative threat with autonomic and inflammatory responses is moderated by stress appraisals and affect. Biol Psychol 2016, 118, 17–24.
We have included a brief background of this complex process in the INTRODUCTION (page 2) according to Reviewer suggestion.
Measurement methods for α-Amylase and DHEA were missed.
ANSWER: Measurement methods were published in a previous report (see Reference 38). In any case, we did a short description in MATERIALS AND METHODS (page 5).
For Fig1 and 2, no significance of the correlation coefficient was provided.
ANSWER: The correlation chart in Figure 1 indicates that the correlation between anxiety and α-amylase is low. A Pearson correlation test returns not significative p-values (>0.05) when comparing the STAI variables (State and Trait) with α-amylase variables (24, 15, 12 and 8h). But it returns significative p-values when comparing the alpha-Amylase variables with each other. With respect DHEA and STAI (Figure 2), a Pearson correlation test returns not-significative p-values (>0.05) when comparing the STAI variables (State and Trait) with DHEA variables (24, 15, 12 and 8h). But it returns significative p-values when comparing the DHEA variables with each other.
Introduction and discussion can be more concise.
ANSWER: Introduction has been shortened in some paragraphs limiting its extension. Discussion contained key information to interpret the Results and this why we did more modest changes.
Reviewer 2 Report
The authors' research objective was to study how particular pattern of stress biomarkers (particularly α-amylase and DHEA) observed in emergency professionals, could influence anxiety, self-efficacy and sleeping quality. However, the research design was not adequate to achieve this objective. The emergency services workplace is a stressful place to work, but as the authors noted, as a result, most of the emergency professionals who participated in this study were a stress-coping group. The anxiety scale was also tested at the start of the working day and not as a result of an acute stress (day job) load. Furthermore, the statistical analysis only analyses the correlations between the anxiety scale and the biomarkers at each measurement point, and does not result in sufficient information for the purpose. The authors conclude that "the hospital's workplace management was good", which is not a conclusion that corresponds to the study objectives. It is desirable to formulate well-considered hypotheses and to develop research designs that can test them.
Author Response
REVIEWER 2
The authors' research objective was to study how particular pattern of stress biomarkers (particularly α-amylase and DHEA) observed in emergency professionals, could influence anxiety, self-efficacy and sleeping quality. However, the research design was not adequate to achieve this objective. The emergency services workplace is a stressful place to work, but as the authors noted, as a result, most of the emergency professionals who participated in this study were a stress-coping group. The anxiety scale was also tested at the start of the working day and not as a result of an acute stress (day job) load. Furthermore, the statistical analysis only analyses the correlations between the anxiety scale and the biomarkers at each measurement point, and does not result in sufficient information for the purpose. The authors conclude that "the hospital's workplace management was good", which is not a conclusion that corresponds to the study objectives. It is desirable to formulate well-considered hypotheses and to develop research designs that can test them.
GENERAL COMMENT
Dear Reviewer, we appreciate your suggestions, which will certainly improve the manuscript. Below we try to answer your questions and concerns.
All changes have also been introduced in the manuscript.
ANSWER: All questionnaires were provided early in the morning at home to each participant. After a short explanation, questionnaires were completed by each participant in a calm environment, usually after work at home. Then questionnaires were picked up at the end of the working day. Therefore, the anxiety scale was tested at the end of the day job, after an acute stress. We have completed this information in the text (page 6).
On the other hand, the anxiety scale gives information regarding the general state during the working day. However, the stress and anti-stress biomarker levels change during the working day, allowing to interpret the general perception of anxiety reflected in the STAI questionnaire. The ideal design would be to pass a STAI questionnaire together to each saliva sample collection, but from a practical and point of view this was not possible.
Reviewer 3 Report
Dear authors,
I read your manuscript with great interest, as stressors during emergency medicine work affects numerous professionals on an every-day basis.
There are some concerns that I have, that in my opinion need to be adressed in order to make the manuscsript more suitable for a broader audience:
-) Abstract (and Overall manuscript): Please use "emergency department" (and the abbreviation ED) instead of "emergency room", as this is more widely accepted terminology.
-) Author list: Was a psychologist or psychiatrist involved? If not, why not? I think this would give the paper a more solid ground of background knowledge.
-) Introduction, "In this context, health professionals ..." - this sentence needs to be cited.
-) Introduction, "It is well established ..." - this sentence needs gramar revising and needs to be cited.
-) Introduction, general: Your introduction is quite long and elaborate. This is not bad per se, but I think in order to keep it interesting for a potential reader, it needs more structure. Maybe introduce bullet points and subheadings, and try to keep it as concise as possible.
-) Methods: How were "normal and stable lifestyle and family habits" defined? This is very vague.
-) Did you make sure that the questionnaires could be answered anonymously? If you fill out a questionnaire about mental health during your lunch break where , let's say, all your collegues are sitting around your, you would probably not admit any problems. This could be a huge bias.
-) Results: You describe statistical differences, but I do not see any tables regarding them. Rather introduce tables with comparisons and respective p-values, instead of tables with little relevant Information like tables 1-4. Your results should read intuitive for a potential reader, right now the section reads a bit confusing.
-) Discussion: Please try to shorten your discussion, introduce subheadings, and find a "flow" of reading and information. This is vital to improve your paper.
-) Please come up with a distinctive Conclusion.
Author Response
GENERAL COMMENT
Dear Reviewer, we appreciate your suggestions, which will certainly improve the manuscript. Below we try to answer your questions and concerns.
All changes have also been introduced in the manuscript.
-) Abstract (and Overall manuscript): Please use "emergency department" (and the abbreviation ED) instead of "emergency room", as this is more widely accepted terminology.
ANSWER: Changes have been performed according to Reviewer suggestions.
-) Author list: Was a psychologist or psychiatrist involved? If not, why not? I think this would give the paper a more solid ground of background knowledge.
ANSWER: Dr Caballero-García has several Masters in Psychology and Health Management. In addition, he is professor in Psychobiology in the Faculty of Psychology at UNED (Spanish University of Distance Education). Alba Roche is Psychologist and Master in Clinical Psychology. Actually, she is doing her doctoral thesis.
-) Introduction, "In this context, health professionals ..." - this sentence needs to be cited.
ANSWER: References 3 and 4 are cited in this sentence (page 2).
-) Introduction, "It is well established ..." - this sentence needs grammar revising and needs to be cited.
ANSWER: Sentence has been revised and changed accordingly: “It is well established that people suffering from intense stress uses to undergo anxiety” (page 3).
-) Introduction, general: Your introduction is quite long and elaborate. This is not bad per se, but I think in order to keep it interesting for a potential reader, it needs more structure. Maybe introduce bullet points and subheadings, and try to keep it as concise as possible.
ANSWER: The Introduction has been shortened and subsections introduced according to Reviewer suggestions (pages 2 and 3).
-) Methods: How were "normal and stable lifestyle and family habits" defined? This is very vague.
ANSWER: In general, a stable lifestyle is considered to be an existence that facilitates physical, psychological and social well-being and maintains favourable conditions to preserve and promote its development. It consists in a physical environment and family relationships that favour the human development of its members and allow them to reach their optimum potential. On the other hand, they have social, physical activity and nutritional habits that allow to maintain the standards of a "healthy life". The definition has been included in the text (page 4).
-) Did you make sure that the questionnaires could be answered anonymously? If you fill out a questionnaire about mental health during your lunch break where , let's say, all your colleagues are sitting around your, you would probably not admit any problems. This could be a huge bias.
ANSWER: All questionnaires were provided early in the morning at home to each participant. After a short explanation, questionnaires were completed by each participant in a calm environment, usually after work at home and not at lunch break in the presence of colleagues. Then questionnaires were picked up at the end of the working day. We have completed this information in the text (page 6).
-) Results: You describe statistical differences, but I do not see any tables regarding them. Rather introduce tables with comparisons and respective p-values, instead of tables with little relevant Information like tables 1-4. Your results should read intuitive for a potential reader, right now the section reads a bit confusing.
ANSWER:
According your suggestions, we have introduced a table with the p-values of the comparisons.
Table 5. p-values for all comparisons for the variables, moment, category and hospital analysed
-) Discussion: Please try to shorten your discussion, introduce subheadings, and find a "flow" of reading and information. This is vital to improve your paper.
ANSWER: Subheadings have been introduced according to Reviewer suggestions (pages 10, 11 and 12).
-) Please come up with a distinctive Conclusion.
ANSWER:” In conclusion, the emergency professionals of the studied hospitals seem to have an adequate work management and the stress controlled during work performance could allow to a correct adaptation to the demanding situations undergone in ED” (page 13).
Round 2
Reviewer 2 Report
The authors' research objective was to study how particular pattern of stress biomarkers (particularly α-amylase and DHEA) observed in emergency professionals, could influence anxiety, self-efficacy and sleeping quality. However, the research design was not adequate to achieve this objective. Although the authors believe that ED work is a stressor, the scale should be tested before and after ED work in order to examine whether stress increases anxiety. The current study design only describes the characteristics of biomarkers in a population with "good hospital workplace management".
Reviewer 3 Report
Dear authors,
Thank you for revising the manuscript, I feel that it has greatly improved. Concerning the psychology background of the authors, I would add this for example in a small part of a sentence in the Methods section, so that potential readers also get this information.
Some additional concerns have come to mind that still need revising in my opinion:
-) Whole manuscript: Minor spelling and grammar errors need to be fixed, for example change "ED" to "EDs" when you mean the plural.
-) Discussion, lines 334-336: This needs to be cited.
-) Discussion, lines 343-346: This needs to be cited (especially the fact about heart rate variability).
-) Discussion: I would add a short "Limitations" section before the conclusion. I would also add a few sentences on what your research / your results now add to the topic (some sort of "pre-conclusion"), and how the information could be used in a daily setting (screen stress levels? etc.).
-) Conclusion: What exactly do you mean by "[...] could allow to correct [...]"? This is very vague. Please give a concise statement that a potential reader can see as a "take home message".